# Prevalence of Multimorbidity in the Middle East: A Systematic Review of Observational Studies

**DOI:** 10.3390/ijerph192416502

**Published:** 2022-12-08

**Authors:** Kalpana Singh, Albara Alomari, Badriya Lenjawi

**Affiliations:** Hamad Medical Corporation, Doha 3050, Qatar

**Keywords:** multimorbidity, prevalence, Middle East, demographics association

## Abstract

Background: There has not been a review that evaluated the prevalence of multimorbidity in the Middle East. This review aims to measure the prevalence, demographic factors, and consequences of multimorbidity in the Middle East region. Study Design: A quantitative systematic review includes cross-sectional and longitudinal cohort studies. Methods: The prevalence systematic review approach from the Joanna Briggs Institute was applied. We searched PsychINFO, MEDLINE, EMCARE, CINAHL, Scopus, Science Direct, and the Cochrane Central Register of Controlled Trials. Data were extracted methodically in accordance with Preferred Reporting Items for Systematic Reviews and Meta-Analyses (PRISMA) standards. Studies written in English and released between 2012 and March 2022 were included. For the meta-analysis, a random-effects model was applied. PROSPERO registration number: CRD42022335534. Results: The final sample consisted of eight cohort and observational studies. The number of participants varied from 354 to 796,427. Multimorbidity was present in all populations with a prevalence of 21.8% (95% confidence interval (CI): 21.7–21.8%). Conclusion: Multimorbidity affects a significant section of the world’s population. A uniform operationalization of multimorbidity is required in the Middle East in order to enable reliable estimates of illness burden, effective disease management, and resource distribution.

## 1. Introduction

Multimorbidity is growing in importance as a public welfare concern as the world’s population rises at an accelerated rate. Patients who have several persistent medical issues frequently experience worse well-being outcomes, such as reduced physical and mental well-being [1], higher mortality rates [2], and frailty [3]. The needs of patients with multimorbidity for restorative care are also pronounced but caring for them places considerable strain on the health system. Rather than a highly specialized but disparate approach used to treat a single disease, patients with multimorbidity require a complex and organized treatment plan [4,5]. This has real implications for disease management, healthcare utilization, and costs [6,7,8].

In order to study the impact of multimorbidity on public welfare and to expand the need for restorative care for patients with multimorbidity, an accurate definition and estimate of its prevalence are crucial [9]. However, the complex nature of multimorbidity poses incredible challenges for research in this area, often incomplete because of irregularities in the conceptualization and definition of multimorbidity [10]. The National Institute for Health and Care Excellence (NICE) has created a multimorbidity guideline and noted that because of the many indicators being employed, determining the prevalence of multimorbidity is difficult [2,11]. Despite these guidelines, there is no universally accepted method for defining and quantifying multimorbidity [12].

The difficulties in defining and quantifying multimorbidity make it challenging to compare different combinations of comorbidities, compare results across populations, and formulate recommendations and interventions. Ambiguity in the conceptualization of multimorbidity leads to the need for a rough agreement on its operationalization [13].

Multimorbidity is more common in high income countries as well as low and middle-income nations (LMIC) (38% vs. 30%, respectively) [14,15] and has been studied in many large reviews [16,17]. However, the majority of Middle Eastern research to date has focused on a single, specific condition or the coexistence of a relatively small number of diseases, such as cancer, diabetes, obesity, and cardiovascular disorders.

The prevalence and presence of multimorbidity have not been investigated in detail in the literature from the Middle East region [18].

Although the urgent need for the introduction of more sustainable multimorbidity care models has finally been acknowledged, politicians and healthcare professionals still want high-quality data to support the argument for change [9]. This review’s objective is to assess the prevalence, demographic influences, and outcomes of multimorbidity in the Middle East. Lebanon, Syria, Iraq, Iran, Palestine, Israel, Jordan, Saudi Arabia, Kuwait, Qatar, Bahrain, United Arab Emirates, Oman, and Yemen are Middle Eastern nations covered by this review.

## 2. Methods

This systematic review followed the Joanna Briggs Institute (JBI) methodology for prevalence systematic reviews [19] and the Preferred Reporting Items for Systematic Reviews and Meta-Analysis (PRISMA) guidelines [20]. A priori protocol has been published in PROSPERO (Registration number CRD42022335534).

### 2.1. Data Sources and Selection Criteria

An electronic search of Medline (Ovid interface), Embase (Ovid interface), Web of Science, and Cochrane Library electronic databases was performed from inception to April 2022. Additionally, Google Scholar, Dissertation Abstracts International, and ProQuest dissertations and theses were searched to find unpublished studies. The term ‘multimorbidity’ and its various spellings (e.g., ‘multi-morbidity’, ‘multimorbidity’s’, ‘multi-morbidities’, ‘multimorbidity’, ‘multi morbidities’, ‘multiple morbidities’, ‘multiple-morbidities’), and ‘prevalence’ or ‘epidemiology’ were used. The Middle East, OR Lebanon, OR Syria, OR Iraq, OR Iran, OR Palestine, OR Israel, OR Jordan, OR Saudi Arabia, OR Kuwait, OR Qatar, OR Bahrain, OR United Arab Emirates, OR Oman, OR Yemen, were also used. The search yielded 282 studies from 2012 to 2022.

This review comprised studies that looked at the prevalence, demographic characteristics, and effects of multimorbidity among adult patients who were at least 18 years old in Middle Eastern nations. Original, peer-reviewed studies were eligible for inclusion (published online, with available abstracts in English). Books, letters, editorials, dissertations/theses, opinion pieces, conference presentations, and abstracts were not included. Only cross-sectional and longitudinal study designs were permitted. Only baseline prevalence was considered when the design was longitudinal. There were no other limitations on the demographic details of the group being studied, such as age, sex, or socioeconomic level. Studies that provided a clear definition of multimorbidity were included, while those that used the terms “comorbidity” and other synonyms were removed.

### 2.2. Search Strategy and Data Extraction

The Endnote X9 bibliographic software was used to compile and upload all recognized citations, and duplicates were eliminated. Those that were not picked up by this function were manually deleted during the initial screening. The first reviewer initially evaluated the titles and abstracts returned by the search (K.S.). To make sure that no eligible research was overlooked, the second reviewer (A.A.) examined a 10% random sample of all references. Studies that met all of the aforementioned eligibility requirements were retained for full-text screening. Two reviewers independently conducted the full-text review (K.S. and A.A.). When there were differences of opinion, K.S. and A.A. talked them out. A consensus was reached to reconcile disagreements. The included studies’ reference lists were manually searched, abstracts were reviewed, and the full text of potentially relevant articles was reviewed.

A sheet for data extraction was created. The following data were taken: study year, study design, study country, data source, sample size, mean age (men/women), definition, measure, and prevalence of multimorbidity, number of diseases, ascertainment of diseases, and combination of diseases.

### 2.3. Quality Assessment

Two reviewers (K.S. and A.A.) independently appraised the included studies using the Joanna Briggs Institute critical appraisal checklist for analytical cross-sectional studies [21]. All studies were included in the review, irrespective of their methodological quality. Any disagreements during the review were resolved through discussion between the authors.

### 2.4. Data Analysis

Estimates were combined using JBI SUMARI, and data were transformed using a random-effects model and the Freeman–Tukey transformation. We measured heterogeneity using the common I^2^ tests. The results are provided in narrative form when statistical pooling was not feasible, with tables and figures to aid in the data presented. The results of overall analyses were compared to multimorbidity operational definitions (2+ disease cutoff points). The meta-analysis includes studies that reported multimorbidity prevalence, both standardized and nonstandardized, using the 2+ chronic diseases cutoff point as the definition. A random-effects model was therefore applied.

## 3. Results

Two hundred eighty-two studies were found through the search. After removing 50 duplicates, 23 studies were identified for possible inclusion, and 232 studies were assessed for relevance using the title and abstract. After the full-text screening, 15 studies were further eliminated, leaving 8 studies for the final quantitative synthesis. The procedure for choosing research is depicted in the PRISMA flow diagram in Figure 1.

### 3.1. Characteristics of the Studies

There were 918310 participants in total among 8 studies (men: 46.5%, women: 53.5%), and the sample sizes varied from 354 to 796,427. The samples ranged in age from 18 years old to 116 years old. Six studies were conducted in Iran [22,23,24,25,26,27], one study from Jordan [28], and one in Qatar [29]. Two studies are cross-sectional, five studies are cohort-design studies, and one study is a retrospective study. The data collection period of the included studies ranged between one to four years. The number of diseases included in examined studies ranged from 3 to 30, with hypertension, diabetes, obesity, cancer, and heart disease, being the five most frequent conditions. Characteristics of the included studies are summarized in Table 1.

Seven studies [22,23,24,25,26,27,29] defined and utilized the term “multimorbidity”. Different methods were employed to identify patients with chronic conditions: self-reports in four studies [24,25,27,28], self-reports and biological measurements in two studies [22,23], and two studies that included two or more chronic conditions in combination [26,29].

The methodological quality of the included studies was moderate. The scores for the methodological quality of the studies ranged from 4 to 9 and indicated that most studies had notable deficiencies (only two, Ahmadi et al. and Ebrahimogli et al., scored 9/9). The median score was 5.5 and ranged from 4/9 to 9/9. Table 1 provides a thorough breakdown of the studies’ caliber (Table 1).

### 3.2. Prevalence of Multimorbidity

The prevalence of multimorbidity varied from 19.4% (95%CI: 19.1–19.8%) to 68.9% (95%CI: 64.0–73.6%). The pooled prevalence of multimorbidity among studies is 21.8% (95%CI: 21.6% to 21.7%) (Figure 2). The leading chronic conditions reported were hypertension, diabetes, obesity, cardiovascular diseases, chronic obstructive pulmonary disease, hyperlipidemia, chronic kidney disease, gastroesophageal reflux disease, cancers, and skin diseases (Table 2).

### 3.3. Multimorbidity and Demographic Factors

All studies explored the multimorbidity burden and how it is associated with age and gender. All studies reported that multimorbidity is associated with increasing age and is more prevalent in females than in males. Three studies reported a significant association between multimorbidity and other factors, such as socioeconomic status, physical activity, education level, and smoking [22,23,24]. Association between literacy and multimorbidity was inconsistent, with two studies finding lower levels of multimorbidity in illiterate people [22,24] and one study reporting a higher level of multimorbidity in illiterate people [25]. In terms of physical activity, patients who have active lifestyles (13.3%) have significantly lower multimorbidity compared with those who have sedentary lifestyles (23.2%) [22,24]. However, one study [25] reported that those who are active have higher (76.1%) multimorbidity compared with those who are not active (33.9%). In terms of socioeconomic status, multimorbidity is inversely associated with economic status, i.e., lower economic status has higher multimorbidity compared with high socioeconomic status, 24.1% and 18.1%, respectively [22,24].

Three studies reported that multimorbidity was positively associated with smoking [22,24,25]. One study [26] explored the combinations of drug therapies used as surrogate markers of chronic conditions.

### 3.4. Association of Multimorbidity

One study [27] reported that the most frequent associations were the lowering of physical functioning disabilities, quality of life, and psychological distress because of multiple comorbidities. In terms of psychological distress, i.e., depression was significantly higher in females (22.4%) compared with males (9.3%).

Both physical and mental component scores were found to be significantly lower in female and educated patients (physical mean score 43.07 vs. 46.54 with *p* = 0.001 and 42.53 vs. 46.82 with *p* < 0.001, and mental mean score 49.80 vs. 52.75 with *p* = 0.022 for education [27].

## 4. Discussion

The majority of research from Middle East countries is focused on a single or specific illness or the coexistence of a relatively small number of diseases, such as cardiovascular ailments, hypertension, diabetes, obesity, and cancer. No review has evaluated the prevalence of multimorbidity in the Middle East. The aim of this review is to measure the prevalence, contributing factors, and associations with multimorbidity in the Middle East region. Despite the extensive search, only eight studies were eligible for inclusion in the review. Evidence from the limited number of studies demonstrated that pooled prevalence of multimorbidity among studies in the Middle East is 21.8%, with substantial variation between studies. This variation may be explained by differences in the definition/measurement of multimorbidity, study populations, demographics, study settings, and self-reported data [30]. Previous studies suggested that the differences in estimated prevalence between countries might also be due to the comparatively limited knowledge of multimorbidity strikes, which consequently led to fewer reports on multimorbidity prevalence [14].

The result of the current review showed that the prevalence of multimorbidity in Middle Eastern countries is lower than the multimorbidity reported in another recent systematic review. The review was conducted among low-income and middle-income countries and reported the Middle East as a region with lower-income countries. The authors reported that the overall prevalence of multimorbidity in the Middle East was 44% [28]. However, their reported prevalence of multimorbidity in the Middle East was based only on two countries; Iran and Jordan [26,28]. The current review reported the findings from three countries (eight studies). The lack of studies and a smaller number of countries reporting multimorbidity in the Middle East highlight the urgency for future studies.

Age and gender are the only demographic factors reported in all studies with consistent results. The included studies reported that increasing age and female patients are significantly associated with multimorbidity. The positive association of multimorbidity with age is consistent with a study comparing low- and middle-income countries with high-income countries [31], showing that increasing age is significantly associated with multimorbidity. Previous studies reported that older age was a potential risk factor for multimorbidity. Most older patients have several chronic diseases and less body fitness to fight multiple diseases, which may be one of the main reasons for fatal outcomes [14,32].

The sex differences in multimorbidity may be associated with context-related indicators for behavioral patterns, such as care seeking, which may influence multimorbidity detection [30]. Where prevalence estimates by gender were reported, females appeared to have higher multimorbidity prevalence rates than males. This is indicative of an association between sex and multimorbidity, which was supported by the findings of previous studies [14,33]. According to research, women have lower socioeconomic status than men in general, which is related to gender inequality and may have a negative impact on health outcomes [34]. Furthermore, according to a recent literature review, women in the Middle East have lower physical activity time due to their busy schedules and home responsibilities, such as caring for extended family members and grandchildren [35].

Cardiometabolic and cardiorespiratory patterns of multimorbidity were the most common, and they share major pathophysiological pathways and risk factors, such as smoking, which may explain why they clustered together [36]. Cigarette smoking is prevalent in Middle Eastern cultures, and they regard smoking as religiously and socially acceptable [37]. The frequent co-occurrence of cardiometabolic conditions and mental disorders among studies in Middle Eastern countries, as demonstrated in this review, is consistent with previous reviews and emphasizes the importance of prevention and management policies addressing environmental and living conditions [38].

## 5. Implications of Findings

Variations in the prevalence of multimorbidity may cause the cost of healthcare, the number of hospital admissions, the distribution of resources, and the overall disease burden to be overestimated or underestimated. As a result, the impacts of health interventions are hampered. Therefore, the necessity for a consistent approach to calculating multimorbidity prevalence grows increasingly important. It is advised that future studies on multimorbidity adhere to a uniform approach, employing a common disease categorization system, disease cutoff point, and multimorbidity measure. Because the age structures of studies vary, prevalence data should be given in both crude and standardized forms. It is important to stratify the findings of prevalence studies by gender and age. When possible, age groupings should be divided into standardized intervals.

The increasing prevalence of multimorbidity in the Middle East indicates that the healthcare system urgently needs to be strengthened in order to handle multimorbidity diagnosis and treatment. According to the evidence available, patients with multiple diseases have significantly higher average outpatient and inpatient visits, resulting in high medical costs. Increased healthcare utilization among multimorbid patients poses challenges for patients, healthcare providers, and the healthcare system.

## 6. Strengths and Limitations

Our rigorous screening and research selection procedures give this review its qualities. Our analysis is the largest systematic review on multimorbidity prevalence to date, as our search technique and inclusion criteria were thorough.

Despite the thorough and systematic search, the main weakness of this review is the lack of available studies representing the different countries of the Middle East, with only three countries, i.e., Iran, Qatar, and Jordan, which was not an exact representation of the Middle East. In addition, the studies included in this review have other limitations, such as age range and gender distribution, or not representing the actual population of the study country, such as focusing on non-camp Syrian refugees in northern Jordan (Rehr et al. [28]) and the Kurdish population of the northwest of Iran (Aminisani et al. [25]). These limitations associated with the included studies may limit the generalizability of the review. More studies are needed to explore multimorbidity in the Middle East region to represent the actual population of the different countries.

Second, all studies are observational studies. Compared with cross-sectional studies, the longitudinal techniques offer a more in-depth understanding of the function of certain risk variables. The third weakness is the fact that almost all studies collected only self-reported data on multimorbidity rather than physical or biochemical information. The self-reported disease is susceptible to recall and self-declaration biases, as well as underreporting or over-reporting of the outcome of interest. The restriction of inclusion criteria to only English-language studies may have also resulted in excluding relevant studies, particularly from Arabic-speaking countries, which may have introduced estimation bias. A causal relationship between the various factors and multimorbidity cannot be established because almost all studies employ cross-sectional designs.

## 7. Conclusions

In spite of the paucity of research, this review’s findings indicate a significant prevalence of multimorbidity in Middle Eastern nations, particularly among women and older patients, with cardiometabolic and cardiorespiratory profiles being the most common types. The estimates of multimorbidity prevalence vary amongst studies. Multimorbidity must be operationalized consistently, as is obvious. Additionally, there is a pressing need for a more extensive epidemiological study on this subject, including the requirement for longitudinal data to access the real direction of multimorbidity and its drivers, to demonstrate causation, and to understand how trends and patterns evolve over time. It will make illness burden estimates more precise, which will improve disease management and resource allocation.

## Figures and Tables

**Figure 1 ijerph-19-16502-f001:**
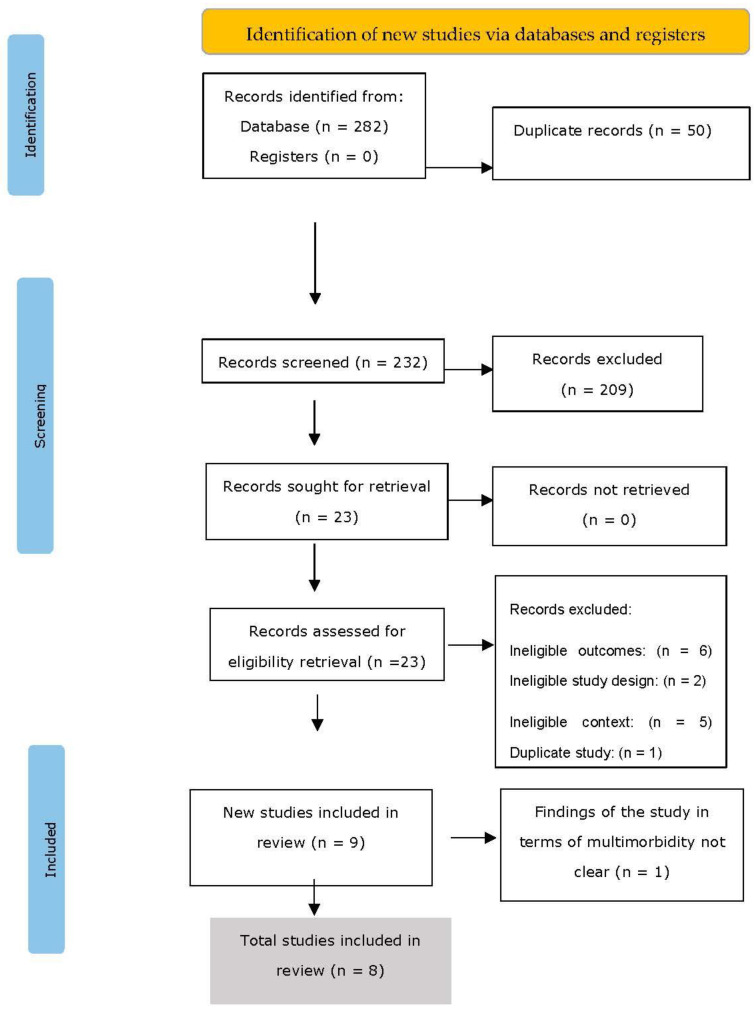
PRISMA (Preferred Reporting Items for Systematic reviews and Meta-Analyses) flowchart of the literature search.

**Figure 2 ijerph-19-16502-f002:**
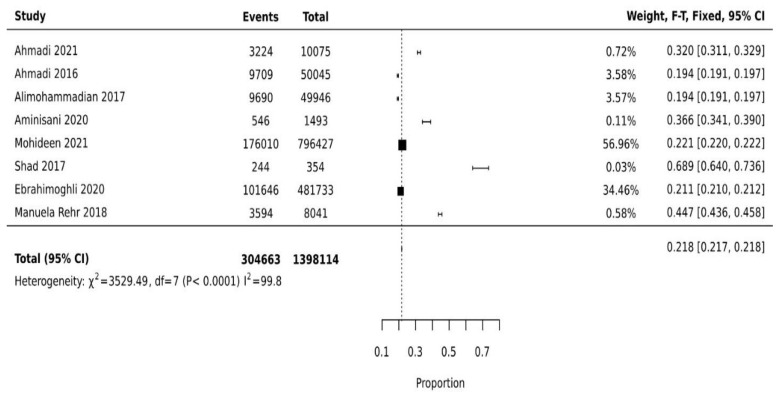
Multimorbidity prevalence in Middle East countries [22,23,24,25,26,27,28,29].

**Table 1 ijerph-19-16502-t001:** Characteristics of included studies.

Study (Country of Study)	Country	Design of Study	Data Collection Period	Sample Size	Age	Gender (% Females)	Prevalence (%) (95%CI)	Quality Assessment
Ahmadi, B. et al., 2016 [22]	Iran	Cohort Study	2004–2008	50,045	40–75 years	58.6%	19.4 (19.1–19.8)	9/9
Ahmadi, A. et al., 2021 [23]	Iran	Cohort Study	2016–2019	10,075	35–70 years	52.80%	32.0 (31.1–32.9)	5/9
Alimohammadian, M. et al., 2017 [24]	Iran	Cohort Study	2004–2008	49,946	40–75 years	58%	19.4 (19.1–19.8)	5/9
Aminisani, N. et al., 2020 [25]	Iran	Cohort Study	2017–2018	1493	50–94 years	62%	36.6 (34.2–39.1)	7/9
Ebrahimoghli, R. et al., 2019 [26]	Iran	Cohort Study	2013–2016	481,733	≥18 years	60.30%	21.1 (21.8–22.2)	9/9
Mohideen, F.S. et al., 2021 [29]	Qatar	Retrospective Cohort Study	2017–2020	796,427	18–116 years	51%	22.1 (22.0–22.2)	4/9
Shad, B. et al., 2017 [27]	Iran	Cross-sectional	2015–2016	354	44–89 years	32%	68.9 (64.0–73.6)	6/9
Rehr, M. et al., 2018 [28]	Jordan	Cross-sectional	2016	2616	≥18 years	55.7%	44.7 (42.4–47.0)	5/9

**Table 2 ijerph-19-16502-t002:** Diseases reported in studies.

Disease	Ahmadi, B. et al., 2016 [22] IRAN	Ahmadi, A. et al., 2021 [23] IRAN	Alimohammadian, M. 2017 [24] IRAN	Aminisani, N. 2020 [25] IRAN	Ebrahimoghli. R. et al., 2019 [26] IRAN	Mohideen, F.S. et al., 2021 [29] QATAR	Shad, B. et al., 2017 [27] IRAN	Rehr, M. et al., 2018 [28]
Chest pain	−	−	−	−	−	−	+	−
Exertional dyspnea	−	−	−	−	−	−	+	−
Obesity	+	+	+	+	+	+	+	−
Asthma	−	−	−	−	−	+		−
Diabetes	+	+	+	+	+	+	+	+
Hypertension	+	+	+	+	+	+	+	+
CKD	+	+	+	+	+	−	−	+
COPD	+	+	+	+	+	−	−	−
CVD	+	+	+	+	+	−	−	+
Gastroesophageal reflux disease(GERD)/Intestinal inflammatory disease/Acid related disorder	+	+	+	+	+	−	−	−
Tuberculosis	+	+	−	+	+	−	−	−
Stroke	+	+	−	+	+	−	−	−
Coronary heart disease	+	+	+	+	+	−	−	−
Liver disease	+	+	−	+	+	−	−	−
Cancer	+	+	+	+	+	−	−	+
Thyroid	−	+	−	−	+	−	−	+
Multiple Sclerosis	−	+	−	−		−	−	−
Hyperlipidemia	−	−	−	−	+	+	−	−
Gout, hyperuricemia	−	−	−	−	+	−	−	−
Migraine	−	−	−	−	+	−	−	−
Glaucoma	−	−	−	−	+	−	−	−
Iron deficiency anemia	−	−	−	−	+	−	−	−
Parkinson diseases	−	−	−	−	+	−	−	−
Bone diseases	−	−	−	−	+	−	−	−
Schizophrenia and bipolar disorders	−	−	−	−	+	−	−	−
Depression, anxiety, and sleep disorders/Eczema/Dementia	−	−	−	−	+	+	−	−
Eczema	−	−	−	−	−	+	−	−
Dementia	−	−	−	−	+	−	−	−

+ means disease is present and − represent the null.

## Data Availability

No new data were created or analyzed in this study. Data sharing is not applicable to this article.

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
