# Peer review of "Prevalence of Multimorbidity in the Middle East: A Systematic Review of Observational Studies"

_ijerph, 2022, doi:10.3390/ijerph192416502_

Round 1
Reviewer 1 Report
The manuscript has many concerns about only using articles with adequate quality. Thus, it was possible to observe the lack of studies in the area in reference to the Middle East.
I consider the manuscript adequate and, therefore, I do not make notes to be corrected.
Author Response
I am thankful to you for review this manuscript.
Thank you
Author Response
Dear Reviewer,
Thank you for your extensive review to improve the manuscript. I have made the changes accordingly. Please see the below mentioned point to point response as follows and edit in the manuscript as an attachment
Prevalence of multimorbidity in middle east: A systematic review and meta-analysis of observational studies
Comment
The reviewed article seeks to summarize the prevalence and importance of multimorbidity in populations of the middle east. There is a clear need for work of this nature, and the manuscript has the potential to provide a meaningful contribution to the literature.
The manuscript has some notable grammatical issues (see below comments), but in general, is well written. The methodologies are clearly explained and are seemingly adequate for the task being performed. The results are provided succinctly and largely satisfactorily (again, see below for specific recommendations).
The greatest deficiencies are in the discussion and conclusion (most specifically, the strengths and limitations). Regardless of the rigor of the review process and meta-analysis procedure, the quality and validity of the study is limited by the available literature for review. These limitations are very notable, and most are not addressed by the authors. Specifically, only eight studies were included (which is acknowledged but not addressed as a weakness). Moreover, six of the studies are derived from Iran, with one each from Qatar and Jordan. The appropriateness of extrapolating results to the middle east is questionable. Indeed, this reviewer is uncertain of the value of attempting to characterize a geographic region with very diverse social, political, economic, and health care environments into a single composite.
Response
Thank you for highlighting the point. We agree with the reviewer with the in terms of geographic region with very diverse social, political, economic and healthcare environment, but the ethnicity of the population is same. Also, we are focusing on the prevalence of multimorbidity which is combination on two conditions.
Lack of studies representing the different countries (or the population) of the middle east region has been addressed in the limitations section as follows:
Despite the thorough and systematic search, the main weakness of this review is the lack of availability of studies representing the different countries of the Middle East, only three countries i.e. IRAN, QATAR and Jordan which was not an exact representation of the Middle East. Also, the studies included in this review have other limitations as age range and gender distribution, or not representing the actual population of the study country such as focused on non-camp Syrian refugees in northern Jordan (Rehr, et al.) and the Kurdish population of the Northwest of Iran (Aminisani, et al.). These limitations associated with the included studies may limit the generalizability of the review. More studies are needed to explore the multimorbidity in the middle east region to represent the actual population of the different countries.
Comment
The next concern builds upon the first- even conclusions related to Iran, Qatar, and Jordan depend upon the representativeness of the sampled populations for the general populace of each of those countries. One reviewed manuscript only examined the Kurdish population of the Northwest of Iran (Aminisani, et al.), and another focused on non-camp Syrian refugees in northern Jordan (Rehr, et al.). Neither of these publications should be presumed to represent the larger population of Iran or Jordan. The remaining studies appear to offer more generalizability publications), but likely still have limitations. These limitations are highlighted by the age ranges and gender distributions described in table 1. Numerous studies had restricted age ranges (40 to 70, 35 to 70, etc.). Therefore, at best, conclusions can only be applied to individuals within those age ranges approximately equals to the country’s population(i.e. the study may have included 40 years old as eligible, but in enrollment required as established medical relationship, it is possible that older individual were over represented). It is clear that the gender distribution typically deviated from the larger population distribution. What selection bias(es) led to this, and how extensive could the impacts be of those bias(es) on the conclusions of the study?
Response:
Thank you for your comment.
These limitations have been added under strengths and limitations section.
Comment
Finally, the entire project is based upon observational studies, which have notable limitations even when ideally designed and executed. As one example, cross-sectional studies provide minimal opportunity for assessing causation. The authors cite Shad, et stating the most frequent consequences (of multimorbidity) were lowering of physical functioning disabilities (line 226). However, given the cross-sectional nature of the study, it is possible that the disabilities preceded the multimorbidities, and may have even contributed to them (for example, a disability may limit physical exercise, which increases risk of diabetes, heart disease, etc.).
Response:
Thank you for your suggestion. We have changed as “One study (27) reported the most frequently associations were lowering of physical functioning disabilities, quality of life and psychological distress (22), it is also possible that the disabilities preceded the multimorbidity, and may have even contributed to them.”
Comment
Only after the above limitations are addressed (geographic representativeness; generalizability, even within the limited countries examined; and observational nature of the work) is it relevant to comment on the self-reporting nature and language limitations (which are both real limitations, but of lesser importance than the above, in my opinion).
Response:
Thank you for your comment. We have changed in line 319 to 328 as follows:
This review main weakness is the geographical region, only three countries i.e. IRAN, QATAR and Jordon which was not exact representation of middle east. Second, all studies are observational studies, comparatively to cross-sectional studies, longitudinal techniques offer more in-depth understanding of the function of certain risk variables.
Comment
Line 51: Strike the apostrophe in multimorbidity and pluralize as multimorbidities. Or better, clarify that you are stating (I believe): ….makes it difficult to compare between different combinations of comorbidities, compare results....."
Response:
Thank you for your suggestion. We have changed accordingly as “The difficulties in defining and quantifying multimorbidity makes it difficult to compare between different combinations of comorbidities, compare results across populations, and formulate recommendations and interventions.”
Comment
Line 54: Multimorbidity is more common in high and low income nations ……. Than? Middle income? The cited reference analyzed low and middle income countries as a single group (LMIC), and the 30% figure cited is in reference to LMIC. Thus, the sentence needs to be changed.
Response:
Thank you for your suggestion. We have done the correction as follows
Multimorbidity is more common in high and low and middle income nations (LMIC) (38% versus 30%, respectively)
Comment
Line 55: “To yet,…….. however ,……..” Is poorly phrased. Consider revising.
Response:
Thank you for your suggestion. We have changed the sentence as “However, the majority of Middle Eastern research to date has focused on a single or specific condition, or on the coexistence of a relatively small number of diseases like cancer, diabetes, obesity, and cardiovascular disorders.”
Comment
Line 95: No hyphen needed for “comor-bidity”
Response:
Thank you for your suggestion. It’s done accordingly.
Comment
Line 129: it is unclear what is meant by the statement “ From 2017 to 2011.” On its face, it implies the search process required five years. I assume that isn’t accurate, but rather that only articles published between these years were examined. If it is correct the criteria should be stated in the material and methods section.
Response:
Thank you for your suggestion. We have changed like “Two hundred eighty-two studies were found through the search.”
Comment
Lines 129 131: There are multiple periods placed within the single sentence describing the numbers of articles identified, with no capitalization after the periods.
Response:
Thank you for your suggestion. Changed accordingly.
Comment
Lines 129 132: It is stated that 23 studies were assessed, resulting in 14 being eliminated. This should leave nine articles, but line 132 states eight remained.
Response:
Thank you for your suggestion. We have corrected accordingly as “After full text screening, 15 studies were further eliminated, leaving 8 studies for the final quantitative synthesis.”
Comment
Table 1: Standardize and improve layout of the table. Problems include (but not limited to):
Response:
Thank you for your suggestion. Table has been updated and modified.
Comment
The first three studies are classified as “Cohort study,” while the next two are simply cohort”. Mohideen, et al. is the only one in parenthesis and is termed a retrospective. Is it a retrospective cohort, a case-control, or some other design?
Response:
Thank you for your suggestion. We have changed this to retrospective cohort.
Comment
- Consider adding punctuation in the sample size figures (i.e., 50,045) for ease of reading. Regardless, eliminate the space in the sample size of Ebrahimoghli et al. study.
Response:
Thank you for your suggestion. We have changed accordingly.
Comment
- Citations should include ‘et al., ‘not at el’.
Response:
Thank you for your suggestion. We have corrected accordingly.
Comment
- Standardize the reporting of 95% CI values: space between the prevalence and opening parentheses for Ahmadi; Ebrahimoghli; and Mohideen; eliminate space after opening parentheses for Alimohammadian; and Rehr; and add opening parentheses for Aminisani.
Response:
Thank you for your suggestion. It’s Done.
Comment
It would appear that the diseases portion of table 1 should be a separate table. Regardless, add ‘D’ in the form of “iseases reported in studies on line 136.
Response:
Thank you for your suggestion. Added D in “iseases
Comment
Table 1 / Table 2 (Disease portion of table):
Response:
Thank you for your suggestion. We did accordingly.
Comment
- Provide a key for all abbreviated diseases. Consider abbreviated gastroesophageal reflux disease (as GERD is commonly used). Is ‘Acid related disorders’ separated fom GERD only because of authors terminology? If both use commonly accepted definitions, they should be combined for analysis.
Response:
Thank you for your suggestion. We have combined accordingly in table2.
Comment
- ordering of disease is confusing. They are not necessarily grouped according to inclusion in various studies, nor are they grouped by systems. Consider reorganizing. Place all G.I conditions together (GERD, intestinal inflammatory dz, acid related disorders), all mental health conditions together, etc.
(placement of eczema between Depression/Anxiety/ sleep disorders and dementia is particularly confusing).
Response:
Thank you for your suggestion. Done Accordingly in Table 2.
Comment
- Line 182 :”…….among 8 research …..” Follow journal standard for reporting numbers however, it would be typical to write out any number less than 10 (i.e. eight). Moreover, it should be among eight publications (or studies).
Response:
Thank you for your suggestion. We have changed accordingly.
Comment
-Line 184: Change 1 to one (see above).
Response:
Thank you for your suggestion. We have changed accordingly.
Comment
Line 187 and 188: change use of the word ‘included’ in one of these instances (…..”number of disease included in examined studies….”)
Response:
Thank you for your suggestion. We have changed accordingly as “. The number of diseases included in examined studies ranged from 3 to 30 diseases; with hypertension, diabetes, obesity, cancer and heart disease, being five most frequent conditions. Characteristics of the included studies are summarized in table 2.
Comment
- Line 191 Change ‘research’ to publication, studies, or articles.
Response:
Thank you for your suggestion. We have changed accordingly.
Comment
Lines 1915- 197: Characterizing quality as ‘average’ is not particularly informative. “Quality assessments (scored from four to nine) found that most studies had notable deficiencies (only two, Ahmadi, et al., and Ebrahimogli, et al. scored 9/9). The median score was …., and ranged from 4/9 to 9/9. Table 1 provides a thorough breakdown of the studies’ caliber”.
Response:
Thank you for your suggestion. We have changed accordingly “The methodological quality of the included studies was moderate. The scores for the methodological quality for the studies ranged from four to nine found that most studies had notable deficiency (only two, Ahmadi, et al., and Ebrahimogli, et al. scored 9/9) The median score was 5.5 and ranged from 4/9 to 9/9. Table 1 provides a thorough breakdown of the studies' caliber.”
Comment
Line211: The statement “Literate people have higher multimorbidity as compared to illiterate” is presented as a definitive conclusion. It is then followed with a diametrically opposite conclusion in lines.
Response:
Thank you for your suggestion. We have modified the sentence as follows “Association between literacy and multimorbidity was inconsistent, with two studies finding lower levels of multimorbidity in illiterate people (23, 24) and one study reported higher level of multimorbidity in illiterate people (25).”
Comment
Line 212-213: It would be better to state “Association between literacy and multimorbidity was inconsistent, with two studies finding lower levels …. And one study finding….”
The same issue with two sentences- the first one is the declarative statement, followed by a contradiction.
Response:
Thank you for your suggestion. Changes are made accordingly “Association between literacy and multimorbidity was inconsistent, with two studies finding lower levels of multimorbidity in illiterate people (23, 24) and one study reported higher level of multimorbidity in illiterate people (25).”
Comment
Line 225: change “most frequently consequences…”to “most frequent associations”
Response:
Thank you for your suggestion. We have changed the sentence to “One study (27) reported the most frequently associations were lowering of physical functioning disabilities, quality of life and psychological distress (22) because of multiple co-morbidities.”
Comment
Line 229: Make score plural (“Both physical and mental component scores were …”). Also, add “to be” between found and significantly.
Response:
Thank you for your suggestion. We changes as “Both physical and mental component scores were found to be significantly lower in female and educated patients (physical mean score 43.07 vs. 46.54 with P = .001 and 42.53 vs. 46.82 with P < .001) and mental mean score 49.98 vs. 52.65 with P = .055 and 49.80 vs. 52.75 with P = .022 for sex and education, respectively (27).“
Comment
Line 231: No statement is made in the materials and methods section regarding a predetermined alpha value for ascribing statistical significance. Convention would assume an alpha of 0.05. The mental scores for women vs. men has a p-value of .055, which would not, in most cases, be considered significant. Yet the preceding sentence says “significantly lower”.
Response:
Thank you for your comment. We have changed as “mental mean score and 49.80 vs. 52.75 with P = .022 for education (27).” Also, we have added the statistical significance in methods section.
Comment
Line 236: “ …. No review evaluating….”
Response:
Thank you for your suggestion. We have changed as “No review has evaluated the prevalence of multimorbidity in the middle east.”
Comment
Line 238: Change “Consequences” to “associations”. The current paper relied exclusively upon observational studies, which were the variable quality. Well-designed and executed cohort studies can be suggestive of consequences. However, given the nature of the totality of the presented evidence, I suggest that the conclusions should be limited to associations rather than suggesting that comorbidities were the causes of some of the outcomes.
Response:
Thank you for your suggestion. We have changed accordingly as “The aim of this review is to measure the prevalence, contributing factors and associations of multimorbidity in the middle east region.
Comment
Line 245: “…. Difference between countries prevalence estimates…” Is possessive (i.e., the countries possess the prevalence estimates. Therefore, there should be an apostrophe after the ‘s’. Alternatively, Phrase it as “…. Differences in estimated prevalence between countries…”
Response:
Thank you for your comment. We have changed the sentence as “Previous study suggested that the differences in estimated prevalence between countries might also be due to the comparatively limited knowledge on multimorbidity strike from which consequently, led to fewer reports on multimorbidity prevalence (31).”
Comment
Line 246: Strike “from’.
Response:Thank you for your comment. We have changed the sentence as “Previous study suggested that the differences in estimated prevalence between countries might also be due to the comparatively limited knowledge on multimorbidity strike from which consequently, led to fewer reports on multimorbidity prevalence (31).”
Comment: Line 249 Change ‘other’ to ‘another’ move the citation for the study you referencing to this point.
Response:Thank you for your suggestion. We have changed accordingly as “The result of the current review showed that prevalence of multimorbidity in the middle eastern countries is lower than multimorbidity reported in another recent systematic review. “
Comment
Line 252-253: The citation provided (29) is for Asogwa, et al. That paper did not report, multimorbidity in the middle east is 44 %” as is characterized in the current work. Rather, the sources they used for Jordan and Iran (Rehr, et al. and Ebrahimoghli, et al., respectively) appear to have been included in the Middle East and North Africa (MENA) geographic classification, for which they reported a prevalence of 33.1%. The only place I see 44% is in the Rehr study, which specifically studies Syrian refugees in Jordan (i.e., it wasn’t an assessment of Jordanians). I would suggest close reexamination of the statements made in these lines.
Response: Thank you for your comment. We have changed citation to Rehr study.
Thank you,
Kalpana

Reviewer 3 Report
It has been a pleasure to have reviewed this paper.
The abstract and introduction are sufficiently significant and representative of the study they precede. The figure showing the schema helps the good understanding of the study scope. The introduction includes the Literature Review, that is correctly founded, but my recommendation would be to split this content and to create a section specifically for Literature Review purpose.
The section corresponding the methodology description of the research is well exposed and well founded. Nevertheless, I recommend to review (and please make a citation) of https://doi.org/10.1007/s11365-022-00800-x where you can see how to describe the process of selection of the type of review (systematic, semi-systematic, etc.). Following the structure and content of that paper, the authors could better explain how they have selected and decided about the methodology to use (systematic review in this case).
The results are clear and define a field of work that more than justifies the conclusions that are advanced. The pictures help the reader to understand, good point!.
The section of limitations has been properly described and exposed.
The conclusions are clear and well exposed, providing a clear picture of the results obtained in the study.
In general, the study is well conducted, well described, and well founded, so I consider the study suitable for publication after improving the methodology section as explained before with the explanation of the path to decide which is the methodology used for the review.
Author Response
Dear Reviewer,
Thank you for giving the time to review the manuscript and given the suggestions.
The abstract and introduction are sufficiently significant and representative of the study they precede. The figure showing the schema helps the good understanding of the study scope. The introduction includes the Literature Review, that is correctly founded, but my recommendation would be to split this content and to create a section specifically for Literature Review purpose.
The section corresponding the methodology description of the research is well exposed and well founded. Nevertheless, I recommend to review (and please make a citation) of https://doi.org/10.1007/s11365-022-00800-x where you can see how to describe the process of selection of the type of review (systematic, semi-systematic, etc.).
Comment
Following the structure and content of that paper, the authors could better explain how they have selected and decided about the methodology to use (systematic review in this case).
Response
This systematic review followed the Joanna Briggs Institute (JBI) methodology for prevalence systematic reviews (20) and the Preferred Reporting Items for Systematic Reviews and Meta-Analysis (PRISMA) guidelines (21)
Thank you,
Kalpana
